# Reprotoxic Effect of Tris(2,3-Dibromopropyl) Isocyanurate (TBC) on Spermatogenic Cells In Vitro

**DOI:** 10.3390/molecules28052337

**Published:** 2023-03-03

**Authors:** Anna Tabęcka-Łonczyńska, Bartosz Skóra, Edyta Kaleniuk, Konrad A. Szychowski

**Affiliations:** Department of Biotechnology and Cell Biology, Medical College, University of Information Technology and Management in Rzeszow, Sucharskiego 2, 35-225 Rzeszow, Poland

**Keywords:** TBC, GC-1 spg cell line, endocrine disruptor, estradiol, reproduction

## Abstract

Tris(2,3-dibromopropyl) isocyanurate (TBC) belongs to the class of novel brominated flame retardants (NFBRs) that are widely used in industry. It has commonly been found in the environment, and its presence has been discovered in living organisms as well. TBC is also described as an endocrine disruptor that is able to affect male reproductive processes through the estrogen receptors (ERs) engaged in the male reproductive processes. With the worsening problem of male infertility in humans, a mechanism is being sought to explain such reproductive difficulties. However, so far, little is known about the mechanism of action of TBC in male reproductive models in vitro. Therefore, the aim of the study was to evaluate the effect of TBC alone and in cotreatment with BHPI (estrogen receptor antagonist), 17β-estradiol (E_2_), and letrozole on the basic metabolic parameters in mouse spermatogenic cells (GC-1 spg) in vitro, as well as the effect of TBC on mRNA expression (*Ki67*, *p53*, *Pparγ*, *Ahr*, and *Esr1*). The presented results show the cytotoxic and apoptotic effects of high micromolar concentrations of TBC on mouse spermatogenic cells. Moreover, an increase in *Pparγ* mRNA levels and a decrease in *Ahr* and *Esr1* gene expression were observed in GS-1spg cells cotreated with E_2_. These results suggest the significant involvement of TBC in the dysregulation of the steroid-based pathway in the male reproductive cell models in vitro and may be the cause of the currently observed deterioration of male fertility. However, more research is needed to reveal the full mechanism of TBC engagement in this phenomenon.

## 1. Introduction

Male infertility is a major global problem that is getting worse every year [1,2,3]. There are many determinants of this phenomenon, inter alia infections, tumors, and environmental pollution. As reported by Bernhard et al. [4], the recently encountered group of important factors includes brominated flame retardants (BFRs), which bioaccumulate and can have a negative effect on living organisms [4,5,6,7,8]. BFRs are very widely used in industry as additives to reduce the level of fire risk in electronic devices, furniture, and textiles [9]. Importantly, BFRs exert an endocrine impact on animals and humans in nontoxic doses [10,11,12]. In the search for safer substances, novel brominated flame retardants (NBFRs) started to be used, butas has been reported many times, their impact is still dangerous to living organisms [9,13,14,15,16]. One of the widely used and little-studied NBFRs is tris(2,3-dibromopropyl) isocyanurate (TBC, also known as TDBP-TAZTO) [17]. This compound was supposed to be an alternative for classic BFRs, e.g., for tetrabromobisphenol A (TBBPA), which exhibits endocrine-disrupting properties, hepatoxicity, immunotoxicity, and neurotoxicity [18,19,20,21]. Unfortunately, current studies have suggested that TBC has similar properties to those of classic BFRs [5,22].

It is well known that estrogens are of great importance for the maintenance of normal physiological functions in the male reproductive system, as they determine steroidogenesis and spermatogenesis in the testes [23]. As shown by Hess, male estrogens are usually produced in the testes or come from local extragonadal production [24]. Based on the available reports, the synthesis of such steroids proceeds with the engagement of the aromatase enzyme (P450arom) through conversion from androgens [25]. On the other hand, a lack or deficiency of estrogens can lead to infertility [26]. Estrogens can act via three receptor subtypes: estrogen receptor alpha (ERα), estrogen receptor beta (ERβ), and G protein-coupled estrogen receptor 1 (GPER1; GPR30) [27]. ERα is a key receptor for male fertility [28,29,30]. To date, molecular docking has shown that TBC has a high affinity for ERα, which may affect the estrogenic pathway [22]. This indicates that TBC has the potential to act as an endocrine disruptor. 

The peroxisome proliferator-activated receptor (PPAR) family is represented by tree subtypes: PPARα, PPARβ, and PPARγ, which are encoded by different genes [31]. All PPARs are important for the regulation of the hypothalamic-pituitary-gonadal axis, and it has been shown that the above-mentioned receptors take part in the gametogenesis process [32]. Interestingly, Liu et al. proved that the inactivation of PPARγ may lead to male fertility disorders [33]. Nevertheless, the PPARγ plays an important role in xenobiotic detoxication, together with the aryl hydrocarbon receptor (AhR), which acts as a multifunctional intracellular sensor [34]. AhR is the transcription factor involved in the metabolism of xenobiotics but is also responsible for many physiological processes, including crosstalk with multiple receptors, including the mentioned PPARγ as well as ERα and RRβ [35]. Conformational changes of the AhR receptor, after ligand binding, lead to the transfer of AhR to the nucleus, and this triggers the expression of AhR target genes, such as cytochrome 1A1 and 1A2, involved in the metabolism of many xenobiotics [36]. Furthermore, AhR is expressed in the testes and can protect the cells of the male reproductive system from external toxins [37]. The AHR/aryl hydrocarbon receptor/nuclear translocator (ARNT) receptor complex facilitates the modification of hormonal signals by influencing the DNA target sites [38]. 

The molecular mechanism of the TBC impact on the male reproductive tract has not been exactly investigated so far. Although there are studies on TBC toxicity in zebrafish (*Danio rerio*) concerning the variety of reproductive and endocrine toxic effects [39], the metabolism of fish is not equal to that of mammals. 

Therefore, the aim of the study was to evaluate the effect of TBC alone and in cotreatment with an ER agonist and antagonist and aromatase on the basic metabolic and apoptotic parameters in mouse spermatogenic cells (GC-1 spg) as a mammalian model in vitro. Moreover, the expression of such key genes as *Ki67*, *p53*, *Pparγ*, *Ahr*, and *Esr1* was measured.

## 2. Results

### 2.1. Metabolic Activity and Caspase-3 Activity of TBC

The first step of the experiment was to investigate the effect of different concentrations of TBC on metabolic activity using the resazurin reduction assay. After 24 h, we did not observe significant changes in the metabolic activity after the treatment of the GC-1 spg cells with TBC (Figure 1A). In turn, after 48 h, we observed a significant 22.75%, 24.48%, and 24.16% decrease in the metabolic activity in the tested cell line after the treatment of the cells with 10, 50, and 100 µM, respectively, compared to the control group (Figure 1A). 

Secondly, after 24 h, we noticed an increase in this caspase-3 activity by 41.96%, compared to the control, in the cells treated with 100 µM of TBC (Figure 1B). In turn, after 48 h, we observed a 39.39% increase in the capase-3 activity, compared to the control, after the treatment with 100 µM of TBC (Figure 1B). 

### 2.2. Cotreatment of the Cells with BHPI, E_2_, and Letrozole

The cotreatment of the cells with the antagonist and the agonist of ER as well as the antagonist of aromatase was performed to determine the engagement of this pathway in the TBC action. After the 48 h exposure of the cells to BHPI (estrogen receptor antagonist), TBC/BHPI, and letrozole, we did not observe any changes in the resazurin reduction test. The groups exposed to TBC, E2, TBC/E_2_, and TBC/letrozole exhibited a 14.67, 24.01, 25.19, and 25.55% decrease in the metabolic activity, respectively, compared to the control (Figure 2A). 

In the case of caspase-3 activity, no changes were observed after the 48 h treatment with TBC, BHPI, TBC/BHPI, letrozole, and TBC/letrozole (Figure 2B). In groups exposed to E_2_ and TBC/E_2_, we observed an 18.26 and 19.62% decrease in caspase-3 activity, respectively, compared to the control (Figure 2B).

### 2.3. Level of mRNA Expression of Ki67, p53, Pparγ, Ahr, Esr1, and Esr2

After the exposure of the GC-1 spg cells to 10 µM of TBC for 24 h, we observed a 21.46% increase in the mRNA expression of *Ki67* compared to the control group (Figure 3A). Similarly, the mRNA expression of *Ki67* significantly increased by 12.51% in the BHPI-treated cells, compared to the control, but decreased by 19.55% in the E_2_-treated cells relative to the control (Figure 3A). Moreover, the effect on the TBC-treated cells was statistically different than in the cells cotreated with TBC and BHPI (19.73% decrease, compared to the TBC alone group) (Figure 3A).

The *p53* mRNA expression level was increased by 22.6% in the TBC-treated cells, compared to the control, and by 42.37% in the cells cotreated with TBC and E_2_, compared to the control (Figure 3B). We also observed a significantly different effect in the cells treated with TBC and those cotreated with TBC and E_2_ (19.75% increase, compared to the TBC alone group) (Figure 3B). 

*Pparγ* gene expression significantly increased in all groups exposed to TBC: TBC alone (42.99%, compared to the control), TBC with BHPI (33.09%, compared to the control), TBC with E_2_ (96.16%, compared to the control), and TBC with letrozole (33.00%, compared to the control) (Figure 3C). In turn, a decrease in the expression of this gene was observed after the use of the other compounds alone, i.e., in the BHPI-(11.44%, compared to the control), E_2_-(24.73%, compared to the control), and letrozole-treated cells (26.72%, compared to the control) (Figure 3C). Moreover, the cells treated with TBC alone and cotreated with E_2_ (53.17% increase, compared to TBC alone) or letrozole (9.99% decrease compared to TBC alone) were characterized by a different effect on the Pparγ gene expression, depending on the treatment used (Figure 3C). 

*Ahr* mRNA expression increased by 13.55% after the treatment of the GC-1 spg cells with BHPI and by 15.15% after the cotreatment with TBC and E_2_, compared to the control (Figure 3D). We also observed a 32.94%, 32.90%, and 18.79% decrease in the expression of this gene in the cells treated with TBC alone, cotreated with TBC and BHPI, and cotreated with TBC and letrozole, respectively, compared to the control (Figure 3D). The TBC-treated cells were characterized by statistically different *Ahr* mRNA expression compared to the TBC- and E_2_- treated cells (38.09% increase, compared to TBC alone) (Figure 3D). 

*Esr1* mRNA expression increased after the treatment of the GC-1 spg cells with TBC (14.77%, compared to the control), BHPI (15.27%, compared to the control), and E_2_ (19.47%, compared to the control) (Figure 3E). In turn, a 24.59% and 17.71% decrease in the level of *Esr1* gene expression was observed in the cells treated with letrozole alone and in the cells cotreated with TBC and letrozole, respectively, relative to the control (Figure 3E). A significant decrease in the *Esr1* mRNA level was also observed after the exposure to TBC with BHPI (20.25%) and TBC with letrozole (32.48%), compared to the TBC-treated cells (Figure 3E). 

No *Esr2* mRNA expression was detected despite the use of two types of primers.

### 2.4. Caspase-3 Protein Expression

After 48 h of the GC-1 spg cells being exposed to TBC, we observed an upregulated level of caspsase-3 protein by 112.44% compared to the control. A similar effect was observed in the cells treated with BHPI (197.05% increase) and cotreated with TBC and BHPI (209.68% increase) compared to the control cells (Figure 4). The decrease in the caspase-3 protein expression was noticed in the cells cotreated with TBC and letrozole compared to the control cells (65.95% decrease) (Figure 4). 

An upregulated level of caspase-3 protein expression (by 97.23%) was also noticed in the TBC and BHPI cotreated cells compared to the TBC variant (Figure 4). In contrast, a decrease in the caspase-3 protein expression was demonstrated in the cells after the exposure to E_2_ (104.76% decrease), TBC with E_2_ (134.16% decrease), letrozole (111.10% decrease), and TBC with letrozole (178.38% decrease), relative to the TBC-treated cells (Figure 4).

## 3. Discussion

The problem of infertility seems to be one of the greatest evolutionary dangers in the modern world [40]. Although the research on female infertility is extensive, the problem of male infertility is relatively poorly understood. The reproductive dysfunction may be caused by toxic substances such as drugs or xenobiotics [41,42]. Our research confirms the reprotoxic effect of TBC on spermatogenic cells (GC-1 spg) in vitro and indicates the possible mechanism of action of this substance in the male reproductive system. In our experiments, after the 48 h exposure, 10, 50, and 100 µM of TBC decreased cell metabolic activity, while the 100 µM concentration increased caspase-3 activity. It is important that the cells used in the present experiments were sensitive to potentially estrogenic active substances. In turn, higher concentrations of TBC have been found to induce apoptosis. To date, a number of papers have shown the toxicity of TBC in a similar concentration range (10 to 100 µM) in different cell types, e.g., the human neuroblastoma (SH-SY5Y) cell line or mouse neurons developing from cerebellar granule cells [43,44]. Interestingly, TBC was nontoxic to mature mouse cerebellar granule neurons and in human hepatocarcinoma (HepG2) cells [43,45]. The proapoptotic properties of TBC have also been described in the SH-SY5Y cell line in the range of 10–100 µM and in the hippocampus of adult mice exposed to 5 or 50 mg/kg of the compound [44,46]. On the other hand, Zhang et al. proved that TBC did not significantly affect the survival and growth properties of zebrafish (*Danio rerio*), which is probably caused by the different metabolism in this model compared to the tested mammalian cells (GC-1spg) [39]. Therefore, our dose-response experiments are consistent with the current state of knowledge.

In the next stage of our study, we decided to determine the molecular mechanism of the action of TBC; to this end, we used an agonist and antagonist of ERs and an aromatase inhibitor. Our experiment showed slight but not statistically significantly intensification of the cytotoxic effect measured by the resazurin reduction test after the treatment of the cells with TBC and E_2_. This suggests that TBC can act in a similar way to E_2_. Moreover, this effect may be the result of the interaction of both substances with ERs. In turn, a decrease in caspase-3 activity was observed in the E_2_, but in the group with cotreatments of E_2_ with TBC, the effect was not enhanced. A similar effect was observed after the application of the ERα receptor antagonist BHPI. However, the decrease in caspase-3 activity intensified in the cells cotreated with BHPI and TBC. In our opinion, this is due to the effect of the tested compounds binding to the ERα active sites and the competition between BHPI, TBC, and E_2_. It was previously found that TBC has an affinity for ERα [22]; therefore, we believe that when Erα active sites are blocked, TBC cannot affect cells, which, in turn, leads to a decrease in apoptosis in GC-1 spg cells. Interestingly, our Western blot analysis of caspase-3 showed an increase in protein expression after TBC or BHPI treatment. Moreover, the level of caspase-3 significantly increased in the TBC/BHPI cotreated group. In turn, E_2_ alone and in the cotreatment with TBC did not change caspase-3 protein expression. Interestingly, letrozole (aromatase inhibitor) in cotreatment with TBC significantly decreased toxicity. The discrepancy in caspase-3 activity and protein expression can be explained by two phenomena. Firstly, the lack of changes in caspase-3 activity and the increase in caspase-3 protein expression in the BHPI-treated groups may have been a result of the activation of autophagic processes in the cells. Moreover, cells can increase the level of native protein, but it may not necessarily be activated. In the case of the E_2_-treated groups, the decrease in caspase-3 activity with the lack of changes in caspase-3 protein expression may have been a result of the decrease in the number of cells measured by the resazurin reduction test. To date, Cao et al. [22] showed that TBC inhibited the proliferation of estrogen-dependent human breast cancer cells (MCF-7/BUS) following E_2_ induction. Given the lack of influence of TBC and E_2_ on the inhibition of the proliferation of the ERα-negative human breast cancer line MDA-MB-231, it was concluded that the mechanism of the antiestrogenic action of TBC involves the ERα pathway [22]. The aforementioned studies also compared the structure of other traditional ERα agonists and antagonists, i.e., E_2_ or 4-hydroxytamoxifen (4OHT, inhibitor of ERs), which proves the possibility of TBC binding to ERα. Moreover, molecular docking and molecular dynamics analysis showed that TBC exhibited a high affinity to the ligand binding domain complex (AF2-ERα LBD) [22]. Therefore, our data suggest the involvement of ERα in the TBC mechanism of action and another undetermined molecular pathway.

In the last part of our study, we tried to elucidate the molecular mechanism of the TBC action. We observed a significant increase in the *p53* gene expression after exposure to TBC. Moreover, after the application of TBC with E_2_, an increase in *p53* expression was observed. Since the p53 protein is involved in the control of the cell cycle, we can assume that TBC inhibits cell division.

This confirms the observed decrease in the cell metabolism/cell number measured by the resazurin reduction test. The expression of the *Ki67* gene, and likewise the *p53* gene, is dependent on the cell growth and cell cycle [47,48] and is responsible for the maintenance of the appropriate DNA structure during mitosis [49,50]. Positive Ki67 expression is involved in the proliferation and differentiation of spermatogonia in the testes and is strictly related to the regulatory function of this protein in this tissue [51,52,53]. Therefore, we used *Ki67* gene expression to evaluate the state of the spermatogenic cell proliferation marker [54]. We observed that the expression of the *Ki67* gene increased both after TBC exposure and after BHPI treatment, which may indicate a cytoprotective cellular response. In contrast, we showed significant downregulation after E_2_ exposure, which confirms the antiproliferative effect of E_2_ on the cells of the male reproductive system [23,55].

There are data suggesting the involvement of both ERα and AhR receptors in the TBC mechanism of action [44]. However, only a few studies show the endocrine-disrupting properties of TBC. Therefore, we decided to investigate the influence of TBC on *Esr1* mRNA expression in the GC-1 spg cells. We observed that the mRNA expression of the *Esr1* gene significantly increased after exposure to TBC, BHPI, and E_2_ alone, which may indicate increased ERα requirement in the cells. On the other hand, when TBC with BHPI was used, the expression of *Esr1* decreased compared to the group exposed to TBC alone. Similarly, letrozole and TBC with letrozole suppressed *Esr1* mRNA expression in the GC-1 spg cells. This may indicate the involvement of other genes in the regulation of the cellular homeostasis mechanism. *Esr2* mRNA expression was also analyzed, but it was not detectable. It is known that letrozole is an inhibitor of cytochrome P450 aromatase (P450arom), which determines the course of the steroidogenesis process [56]. P450arom is an enzyme involved in the proper course of many physiological processes in various tissues, including gonads [57]. It is mainly responsible for the irreversible bioconversion of androgens to estrogens and for the androgen/estrogen balance in the body [58]. 

It is well known that flame retardants may act via multiple steroid and nuclear receptors [59]. Therefore, we examined the mRNA expression of the *Pparγ* and *Ahr* genes. PPARγ regulates the level of expression of lipid metabolism-related genes in Sertoli cells after incubation with a PPARγ activator [60]. The activation of PPARγ in Sertoli cells ensures the proper course of energy metabolism in both Sertoli and germ cells and is responsible for the production of lactate, which is necessary as a metabolic substrate for the functioning of germ cells [61,62]. Studies on mouse Leydig cells have shown that the regulation of the steroidogenesis pathway is mediated by PPARγ-G-protein coupled estrogen receptor (GPER) interaction [63]. Moreover, the activation of PPARγ regulates Leydig cell morphology, which is of the greatest importance for steroid biosynthesis [63]. Therefore, PPARγ is a factor that controls the normal course of male reproductive functions, as it has been shown that the activation of PPARγ nuclear receptors participates in the suppression of steroidogenesis by phthalates [64]. 

It is worth adding that the research conducted by Hamers [65] showed the relationship between the dose of BFR used and the interaction with the androgen, estrogen, or progesterone pathways [65]. Other studies have found that brominated TBBPA and chlorinated tetrachlorobisphenol A (TCBPA) activate PPARγ [66]. It has also been observed that the greater the level of bromination of bisphenol A (BPA) analogs, the greater their ability to activate PPARγ and the lower their estrogenic potential. It has, therefore, been suggested that these compounds may act as agonists and, through their influence on PPARγ, they determine many of the body’s functions that are regulated by this enzyme [67]. In addition, other studies have shown that the activation of PPARγ induces the expression of antioxidant enzymes, e.g., catalase and superoxide dismutase (SOD), i.e., two enzymes capable of alleviating oxidative stress and inhibiting NADPH oxidase [68,69,70]. 

Our results showed an increase in the expression of *Pparγ* mRNA in all TBC exposure groups, with the highest level noted after the cotreatment of the cells with TBC and E_2_. Hence, we suppose that Pparγ may have cytoprotective properties in the tested cells representing the male reproductive system in vitro. El-Sayed et al. [71] confirmed the antioxidant activity of Pparγ in rat testes tissue [71]. However, cell protection may limit the normal functions performed in the testes and lead to reduced or complete infertility as a consequence of the response to TBC. We also observed that the mRNA expression of *Pparγ* increased after exposure to letrozole with TBC, although *AhR* and *Esr1* mRNA expression decreased. Zhang et al. showed that the decrease in the *Esr1* gene expression level is strictly related to letrozole sensitivity in human breast cancer cells [72]. Moreover, the crosstalk between Pparγ, AhR, ERα, and ERβ receptors has been described [73,74]. Therefore, we believe that an increase in *Pparγ* gene expression causes a decrease in *AhR* mRNA expression, which is similar to previously published data concerning other types of xenobiotics [74]. On the other hand, Nilsson et al. [50] proved that the knockout of PPARγ by respective siRNA causes an increase in *ERα* gene expression and activity [75]. As shown by Rezvanfar et al., in mice with letrozole-induced hyperandrogenization, an increase in *Pparγ* gene expression may occur, resulting in the protective effect of this protein [76]. Therefore, we cannot exclude that TBC can affect all the aforementioned receptors, i.e., *Pparγ*, *AhR*, and *ERs*, in our experimental model.

As reviewed by Bar et al. [17], many studies postulate some correlation between ERα and intracellular receptors, such as AhR, in the context of TBC [17,74]. Nonetheless, there are no empirical studies that prove the aforementioned phenomenon in the context of the TBC. However, our data show a significant increase in *Ahr* mRNA expression in the spermatogenic cells after exposure to TBC with E_2_. Such activation may represent the fragmentary defense/protective mechanism of cells against toxins, but it may also be conducive to male reproduction problems. Studies conducted on rat and human testes confirm the negative effect of environmental dioxins on the cells of the male reproductive system [77]. The effect of 2,3,7,8-tetrachlorodibenzo-p-dioxin (TCDD) on the testes has indicated that this organ is extremely sensitive to toxins [38]. In addition, it directly affects the control of cell death during the spermatogenesis process by activating AhR and inducing apoptosis in target cells [78]. Even short-term exposure disrupts the signaling between Sertoli cells and germ cells, which, in turn, has an impact on the spermatogenesis process. The AhR/ARNT receptor complex was able to modify hormonal responses directly by influencing DNA [38]. It has also been shown to reduce the volume of Leydig cells [79]. Further studies confirmed the regulatory function of AhR on the Notch signaling pathway (ligand-, receptor-based signaling pathway) by maintaining contact between adjacent cells [80,81]. The results of these studies may suggest the mechanism of male infertility [42,81]. 

## 4. Materials and Methods

### 4.1. Chemicals

The chemicals were of analytical grade and were used without further chemical modification. Unless otherwise stated, the chemicals and reagents were purchased from Sigma-Aldrich (Poznan, Poland), Thermo Fisher Scientific (Waltham, MA, USA), or Cayman Chemicals (Ann Arbor, MI, USA). 

### 4.2. Cell Culture and Treatment

The mouse-derived spermatogonia (GC-1 spg) cell line was obtained from the American Type Culture Collection (ATTC, CRL-2053, Manassas, VA, USA). The GC-1 spg cell line was maintained in Dulbecco’s Modified Eagle’s Medium (DMEM) without phenol red (10–013-CVR), supplemented with charcoal/dextran-treated 10% fetal bovine serum (FBS) with 100 U/mL of penicillin, 0.10 mg/mL of streptomycin, and 250 ng/mL of amphotericin B. The cells were maintained in standard conditions (37 °C in a humidified atmosphere and 5% CO_2_) and passaged by trypsinization every three days (confluency ~80%). For the experiments, the cells were seeded in 96-well (resazurin reduction test), 12-well (Real-Time PCR analysis), or 6-well culture plates (Western Blot measurement) at a density of 4 × 10^4^ cells/well, 1 × 10^5^ cell/well, or 1.2 × 10^5^ cells/well, respectively, and then initially cultured for 24 h. In the subsequent dose-response experiments, the medium was replaced with fresh medium containing increasing concentrations (1, 10, 50, and 100 nM and 1, 10, 50, and 100 µM) of TBC for 48 h, unless otherwise stated. In the case of the cell cotreatment, the medium was enriched with 10 µM of TBC (selected on the basis of dose-response data) and 10 µM of 1,3-dihydro-3,3-bis(4-hydroxyphenyl)-7-methyl-2H-indol-2-one (BHPI—Erα antagonist), 100 nM of estradiol (E_2_—ER agonist), or 1 µM of letrozole (aromatase inhibitor) for 48 h. 

### 4.3. Resazurin Reduction Assay

The resazurin reduction test was used to determine the metabolic activity of the cells. Metabolically active cells have the ability to convert blue resazurin (non-fluorescent form) to red resorufin (fluorescent form) [82]. The cells were seeded in 96-well plates and incubated with different concentrations of TBC for 48 h at 37 °C. Then, the medium was removed, and new medium containing 1% of FBS and 10% of resazurin (100 µL) was added. After 30 and 60 min. of incubation at 37 °C, the fluorescence level was measured using a microplate reader FilterMax F5 Multi Mode (Molecular Devices, Corp., Sunnyvale, CA, USA) at 530 nm excitation and 590 nm emission wavelengths. 

### 4.4. Caspase-3 Activity

Caspase-3 activity as an apoptosis marker for GC-1 spg cells was measured according to the protocol proposed by Nicholson et al. [83]. After the incubation with different concentrations of TBC and the cotreatment with the tested compounds, the cells were lysed using CAB buffer (50 mM HEPES, pH 7.4, 100 mM NaCl, 0.1% CHAPS, 1 mM EDTA, 10% glycerol, and 10 mM DTT) at 10 °C for 10 min. Then, the caspase-3 substrate (Ac-DEVD-pNA) was added. Subsequently, the absorbance was measured (405 nm) for 30 min using a microplate reader (FilterMax F5 Multi-Mode). 

### 4.5. Western Blotting

The caspase-3 protein level was visualized by Western blotting, according to Szychowski et al. [84], with modifications. Briefly, the GC-1 spg cells were seeded and treated as described above for 48 h. Subsequently, the medium was removed, and the cells were washed once with phosphate-buffered saline without Ca^2+^ and Mg^+^ (PBS) and lysed using the ice-cold radioimmunoprecipitation assay (RIPA) buffer supplemented with protease inhibitors. Next, the protein concentration was determined by the Bradford quantitation assay using bovine serum albumin (BSA) as a standard. Next, 30 µg of the samples were fractionated by 10% SDS-PAGE and electrotransferred from polyacrylamide gel to PVDF membranes. Unspecific protein-binding sites were blocked using 1% BSA in TBST and incubated with the primary antibodies overnight at 4 °C: anti-β-actin (1:10,000; #PA1-16889, RRID: AB_568434) and anti-caspase 3 (9H19L2) (1:1000, #PA1-700182; RRID: AB_2532293). Next, the membranes were washed four times and incubated (1h, RT) with secondary HRP-conjugated antibodies: anti-mouse (1:40,000; #A9044, RRID: AB_258431). Subsequently, the membranes were washed three times with TBST and visualized by the chemiluminescent substrate (ECL) using the Western Blotting Luminol Reagent (Santa Cruz Biotechnology, Inc., Dallas, TX, USA) and LiCor C-DiGit according to the provided instructions. The densitometric analysis was performed with the GelQuantNET software. The bands were quantified and normalized to their corresponding β-actin bands (loading control).

### 4.6. Real-Time PCR

The Real-Time PCR method was chosen to determine the impact of TBC and the tested compounds on the expression of certain genes, as in Skóra and Szychowski [85]. After the 24 h exposure to 1 µM of TBC alone or in the cotreatment with the other compounds, RNA was extracted using the Universal RNA Purification Kit according to the manufacturer’s instructions (EURx, Gdańsk, Polska). Then, the RNA quality and quantity were controlled spectrophotometrically at 260 and 280 nm and normalized to 500 ng (ND/1000 UV/Vis; Thermo Fisher NanoDrop, Waltham, MA, USA). Subsequently, the reverse transcription reaction (RT) was performed using the obtained RNA as a template and the High-Capacity cDNA Reverse Transcription Kit according to the producer’s manual (Thermo Fisher, Waltham, MA, USA). Next, the obtained RT reaction products were amplified using Fast Probe qPCR Master Mix (2x), plus ROX Solution (EURx), primers, and TaqMan probes specific for genes *Gapdh* (Mm99999915_g1), *Ki67* (Mm01278617_m1), *p53* (Mm01731290_g1), *Pparγ* (Mm00440945_m1), *Ahr* (Mm01291777_m1), *Esr1* (Mm00433147_m1), and *Esr2* (Mm01281854_m1 and Mm00599821_m1). The total volume of the reaction was 20 µL. The qPCR reactions were conducted according to the standard procedures: 2 min at 50 °C and 10 min at 95 °C, followed by 40 cycles of 15 s at 95 °C and 1 min at 60 °C. The threshold value (Ct) for each sample was set during the exponential phase, and the ΔΔCt method was used for data analysis. *Gapdh* was used as a reference gene. 

### 4.7. Statistical Analysis

All data are presented as means ± standard deviation (SD) of three independent experiments (total number of replicates *n* = 9). The results were analyzed with a one-way analysis of variance (ANOVA) followed by posthoc Tukey’s or t-test using GraphPad Prism 8.0 Statistical Analysis Panel. Significant differences were marked as follows: *** *p* < 0.001, ** *p* < 0.01, and * *p* < 0.05 vs. the control group. # *p* < 0.05 vs. the TBC-exposed group.

## 5. Conclusions and Perspectives

In conclusion, the presented research shows that TBC, as a ubiquitous environmental pollutant, has a toxic effect on spermatogenic cells (GC-1 spg cells). In addition, acting together with E_2_, it has a strengthening and additive effect, activating steroid and nuclear receptors, e.g., Pparγ and AhR. Moreover, given the comparison of TBC to BHPI and E_2_ and the measurement of gene expression, we believe that TBC is at least a weak estrogen and competes with E_2_ for cellular receptors. These results suggest the tremendous effect of TBC on the cells of the male reproductive system. However, the exact mechanism still remains to be clarified; therefore, we plan to conduct studies on human cell lines and animals for the next stages of this research.

## Figures and Tables

**Figure 1 molecules-28-02337-f001:**
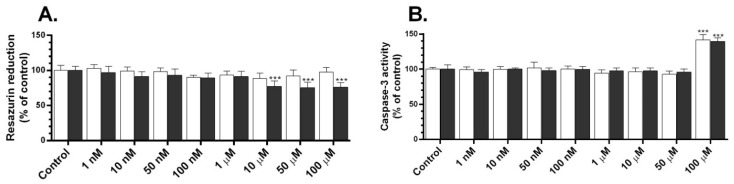
Effect of 1, 10, 50, and 100 nM and 1, 10, 50, and 100 μM of TBC on the level of resazurin reduction (**A**) and caspase-3 activity (**B**) in GC-1 spg cell line after 24 (white bars) and 48 (black bars) h exposure. Data are expressed as means ± SD of three independent experiments, each conducted in six replicates per treatment group. *** *p* < 0.001 vs. the vehicle control.

**Figure 2 molecules-28-02337-f002:**
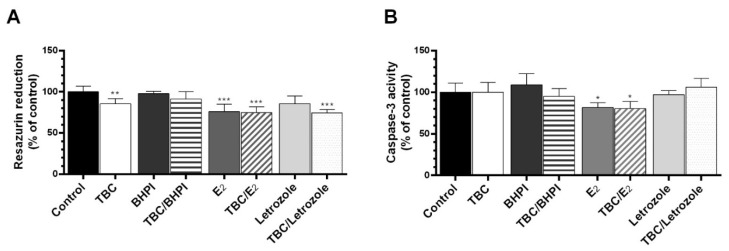
Effect of 10 µM of TBC and TBC in the cotreatment (BHPI—10 µM, E_2_—100 nM, letrozole—1 µM) on the level of resazurin reduction (**A**) and caspase-3 activity (**B**) in the GC-1 spg cell line after 48 h exposure. Data are expressed as means ± SD of three independent experiments, each conducted in six replicates per treatment group. * *p* < 0.05, ** *p* < 0.01, *** *p* < 0.001 vs. the vehicle control.

**Figure 3 molecules-28-02337-f003:**
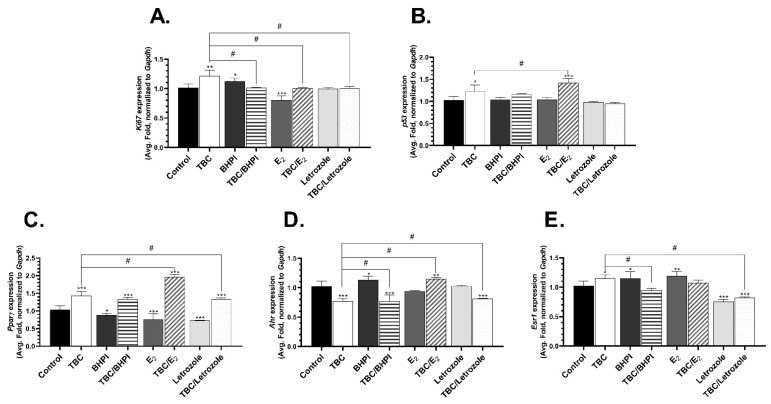
Effect of 10 μM TBC and TBC in the cotreatment (BHPI—10 µM, E_2_—100 nM, letrozole—1 µM) on the *Ki67* (**A**), *p53* (**B**), *Pparγ* (**C**), *Ahr* (**D**), *Esr1* (**E**), and *Gapdh* mRNA expression in the GC-1 spg cells after 48 h exposure. The mRNA expression was normalized to *Gapdh*. Data are expressed as means ± SD of three independent experiments, each conducted in six replicates per treatment group. * *p* < 0.05, ** *p* < 0.01, *** *p* < 0.001 vs. the vehicle control; # *p* < 0.05 vs. cells treated by TBC alone.

**Figure 4 molecules-28-02337-f004:**
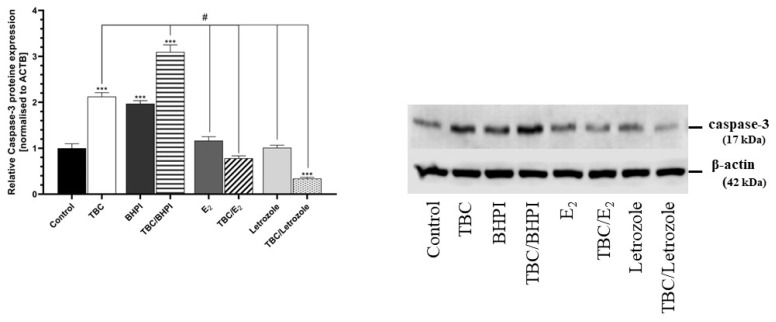
Effect of 10 µM TBC and TBC in the cotreatment (BHPI—10 µM, E_2_—100 nM, letrozole—1 µM) on the caspase-3 protein expression level in the GC-1 spg cell line after 48 h exposure. Data are expressed as means ± SD of three independent experiments, each conducted in six replicates per treatment group. *** *p* < 0.001 vs. the vehicle control; # *p* < 0.05 vs. cells treated by TBC alone.

## Data Availability

The data will be available upon request from the corresponding author.

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
