# Peer review of "Reprotoxic Effect of Tris(2,3-Dibromopropyl) Isocyanurate (TBC) on Spermatogenic Cells In Vitro"

_molecules, 2023, doi:10.3390/molecules28052337_

Round 1
Reviewer 1 Report
This manuscript reports on the reprotoxic effect of TBC(which was considered as an alternative to TBBPA) on spermatogenic cells in vitro. Study on the effect of TBC and in co-treatment with BHPI, E2, and letrozole on basic metabolic parameters as well as mRNA expression is interesting and meaningful. The results and conclusions are basically reliable and reasonable, although there is a lack of research on the endocrine disrupting mechanisms of TBC. In summary, I agree that this article can be accepted after minor revision. This is a good article.
1. It seems that the chapter on "Materials and methods" should be section of 2.0.
2. Some readers may not be aware of BHPI, please clarify when it first appears in the abstract or main text.
3. P7 Line242 "Moreover, our results show that TBC acts through ERs affecting steroidogenesis." Which part of the previous study confirms this?
4. In the discussion section, it is suggested that some recent research literature from the last three years be added to expand the discussion.
Author Response
Response to the Reviewer #1’s comments
- It seems that the chapter on "Materials and methods" should be section of 2.0.
Response:
We want to thank the Reviewer #1 for this suggestion, we have read the manuscript once again however according to the requirements of the Molecules journal, the Material and Methods chapter should be placed after the discussion. That is why it is number 4 in our manuscript. This is an individual editorial requirement that we have tried to fulfill
- Some readers may not be aware of BHPI, please clarify when it first appears in the abstract or main text.
Response:
Thank you for your right attention. Information about full-name of BHPI is provided in the text of the manuscript (lines 18 (abstract) and 105 (main text of manuscript)).
- P7 Line242 "Moreover, our results show that TBC acts through ERs affecting steroidogenesis." Which part of the previous study confirms this?
Response:
We have decided to remove this sentence as it is unjustified and too speculative. Further studies are being performed to determine in detail the afore-mentioned statement, which will be published in our next paper.
- In the discussion section, it is suggested that some recent research literature from the last three years be added to expand the discussion.
Response:
We are very grateful for the Reviewer’s suggestion, unfortunately data concerning TBC are extremely limited. We have found some errors in the literature, and the whole list has been corrected.
Reviewer 2 Report
Dear Athors, please corrected the marked sentences!
Dear Editor, the mansucript with the title "Reprotoxic effect of tris(2,3-dibromopropyl) isocyanurate (TBC) on spermatogenic cells in vitro" is very well written and describes the influence of this toxic substance coming from pollution of the environment in the male reproductive cells.
The influence of TBC in male reproductive cells as a model of research!The male infertility is very important in human beings as well in animals (e.g. stallions, bulls, ...).This manuscript describes the effect of TBC, TBC with co-treatment 17 BHPI, 17β-estradiol, and letrozole in mouse spermatogenic 18 cells.
After correction I shall recommend the acceptance of this manuscript for publication.

Author Response
Response to the Reviewer#2’s comments
After correction I shall recommend the acceptance of this manuscript for publication.
Response:
Thank you for your attention, the manuscript has been carefully reviewed and revised by a native speaker. If it is required, we can send confirmation of the correction.
Reviewer 3 Report
In this paper, the author studied the reprotoxic effect of tris(2,3-dibromopropyl) isocyanurate (TBC)on spermatogenic cells in vitro. It is widely known that TBC is a typical hexabromoheterocyclic brominated flame retardant with stable chemical properties. At present, TBC is added to the production of polyolefins, polyvinyl chloride, polyphenylene, synthetic rubber, optical fiber, etc. The test results of relevant environmental samples show that TBC has environmental durability and bioaccumulation. It is of scientific and practical significance to investigate the endocrine disrupting effect of this compound.
Although the topic has certain significance, the research design is relatively poor, and the existing experimental results cannot support the research conclusion. In a word, the author's research provides us with some possibilities, but it is still a long way from clarifying the possible mechanism of action of TBC.
In addition, there are some obvious mistakes in this paper.
Example 1: The results in Figure 1 and Figure 2 are inconsistent.
Figure 1: Compared with the control group, there was a significant difference in the level of diazurin reduction after exposure to 10 uM TBC for 48 h.
Figure 2: Compared with the control group, there was no significant difference in the level of diazurin reduction after exposure to 10 uM TBC for 48 h.
Example 2: The title of Part 2.3 is inconsistent with that of Figure 3.
2.3. Level of mRNA expression of Ki67, p53, Pparγ, Ahr, Esr1, and Esr2
Figure 3. Effect of 10 μM TBC and TBC in the co-treatment (BHPI-10 µM, E2-100 nM, letrozole-1 µM) on the KI67, Bp53, Pparγ, Ahr, Esr1, and Gapdh mRNA expression in the GC-1 spg cells after 48 h exposure.
Author Response
Response to Comments from Reviewer #3
# Although the topic has certain significance, the research design is relatively poor, and the existing experimental results cannot support the research conclusion. In a word, the author's research provides us with some possibilities, but it is still a long way from clarifying the possible mechanism of action of TBC.
Response:
We are extremely grateful for the review of the manuscript and showing very vestigial attention to some problems. Our study is preliminary and ongoing, however, due to quite significant findings, we wanted to share it quickly. On their basis, further analyzes will be carried out, which will allow for a detailed study of the effect of TBC on reproductive cells. So far, no one has studied the effect of TBC on male reproduction, and these discoveries may be crucial in today's world, in the face of increasingly common male infertility. In addition, this work will form the basis for the implementation of further planned stages of research in this direction, and will be the first to be valuable and highly cited. We agree that the paper does not explain all the questions that arise in the implementation of a new scientific problem, therefore this topic will be developed and continued.
However, to make the manuscript less speculative and more readable for the future readers we have decided to delete sentence which was unjustified and do not supported by the data.
#In addition, there are some obvious mistakes in this paper.
Example 1: The results in Figure 1 and Figure 2 are inconsistent.
Figure 1: Compared with the control group, there was a significant difference in the level of diazurin reduction after exposure to 10 uM TBC for 48 h.
Figure 2: Compared with the control group, there was no significant difference in the level of diazurin reduction after exposure to 10 uM TBC for 48 h.
Response:
We sincerely apologize for the presented mistake. Figure 2A lacks an asterisk, which reflects a significant difference between control and TBC for 48 h. The figure 2A has been revised and now shows the statistically significant difference seen in Figure 1A, and the results are consistent now.
Error occurred during downloading data from the microplate reader - wrong data file downloaded. Data are corrected now for which we thank the Reviewer.
Example 2: The title of Part 2.3 is inconsistent with that of Figure 3.
2.3. Level of mRNA expression of Ki67, p53, Pparγ, Ahr, Esr1, and Esr2
Figure 3. Effect of 10 μM TBC and TBC in the co-treatment (BHPI-10 µM, E2-100 nM, letrozole-1 µM) on the KI67, Bp53, Pparγ, Ahr, Esr1, and Gapdh mRNA expression in the GC-1 spg cells after 48 h exposure.
Response:
The discrepancy in the name of chapter 2.3 has been corrected - the names of the presented genes have been changed in the description to Figure 3. Moreover, gene symbols are italicized. Protein symbols are not italicized. Proteins and human genes are written in capital letters. Animal genes and proteins written with only the first capital letter.
Moreover, discrepancy in activity and expression have been described. “Firstly, the lack of changes in caspase-3 activity and the increase in caspase-3 protein expression in the BHPI-treated groups may have been a result of activation of autophagic processes in the cells. Moreover, cells can increase the level of native protein but it may not necessarily be activated.”
Round 2
Reviewer 3 Report
Personally, the quality of the first draft can reflect the author's rigor and the reliability of the experimental data. After carefully reviewing the author's reply letter and revised draft, I still adhere to my opinion of first instance.

Author Response
Thank you very much for any suggestions. We wish to use any advice to improve the manuscript and hope that our answers and corrections in the text of the manuscript will satisfy the reviewer.
# The authors mentioned that - So far, no one has studied the effect of TBC on male reproduction. As far as I know, the following article is the first data describing the effects of TBC in vivo. Zhang X, Li J, Chen MJ, Wu L, Zhang C, Zhang J, Zhou QF, Liang Y. Toxicity of the brominated flame retardant tris-(2,3-dibromopropyl) isocyanurate in zebrafish (Danio rerio) Chinese Science Bulletin. 2011;56:1548–1555. doi: 10.1007/S11434-011-4471-6
Response:
So far, the effect of TBC on the cells of the male reproductive system of mammals has not been studied! Our research relates to the mechanism of action of this substance on a mouse model, but in the longer term in relation to humans. Metabolism of fish is not equal to that of mammals, so we didn't take fish into account. This is the only data from fish that is completely different from mammals. However, the exact mechanism of action on the reproductive cells of male mammals is unknown, which may be crucial for improving the situation of male infertility. Nevertheless, we agree with the Reviewer that adding extra paragraphs in our study would be beneficial, therefore we have added the following excerpt in the text of the manuscript in the Introduction and Discussion sections :
L75-82:
“The molecular mechanism of the TBC impact on the male reproductive tract has not been exactly investigated so far. Although, there are studies of TBC toxicity on zebrafish (Danio rerio), concerning the variety of reproductive and endocrine toxic effects, (39) the metabolism of fish is not equal to that of mammals.
Therefore, the aim of the study was to evaluate the effect of TBC alone and in co-treatment with an ER agonist and antagonist and aromatase on the basic metabolic and apoptotic parameters in mouse spermatogenic cells (GC-1 spg) as a mammalian model in vitro.”
And
L211 – 214:
“On the other hand, Zhang et al. (39) have proved that TBC did not affect significantly the survival and growth properties in the zebrafish (Danio rerio), which is probably caused be different metabolism in this model than in tested mammalian cells (GC-1spg) (REF).”
# With regard to the toxic effects of this compound, the following review article is representative. Bar M, Szychowski KA. Comprehensive review of the impact of tris(2,3-dibromopropyl) isocyanurate (TBC or TDBP-TAZTO) on living organisms and the environment. Environ Geochem Health. 2022 Dec;44(12):4203-4218. doi: 10.1007/s10653-022-01206-y. Epub 2022 Feb 1. PMID: 35103871; PMCID: PMC9675702.
# At the end of this review article, the author concluded that - This review is the first summary of the current state of knowledge of the presence of TBC in environmental samples and living organisms as well as the mechanism of its action in cells. Currently, TBC, which was supposed to be an alternative to classic BFRs, turns out to be an equally serious problem for the environment and living organisms. TBC is widely detected in soil, sediments, river water, and such materials as microplastic, curtains, and e-waste devices. This compound has strong potential to accumulate in organisms and has been detected in tissues of a number of species; for instance, it can easily penetrate the brain. According to the presented literature, TBC causes damage mainly to the nervous and endocrine systems, lungs, and liver and impairs the function and development of the reproductive system. To date, scientists agree that TBC acts with involvement of ERα and AhR receptors. Moreover, due to the known cross talk between these receptors, their interaction cannot be excluded in the TBC mechanism of action. Most importantly, since TBC interferes with the correct expression and/or activity of P450 enzymes, it may interfere with the proper metabolism of other xenobiotics. Given the many disadvantages of TBC and its accumulation in living organisms affecting their function, we propose that new and safe alternatives to BFRs should be sought.
Response:
In response to the Reviewer, in review published by our team (Bar et al.) “scientists agree that TBC acts with involvement of ERα and AhR receptors” – but this is only hypothesis never tested - which is based on a well-described cross talk between these receptors. Our message is to learn the exact mechanism of action of TBC, especially within the male reproductive system in an in vitro mouse model. On this basis, we plan to conduct further research that will allow us to learn the detailed metabolic pathways of TBC activity on the reproductive cells of males. Molecular analysis seems to be necessary to study the effect of TBC on this particular type of cells, which will allow to isolate the mechanism of its action as an endocrine disruptor (detailed analysis on the path of formation of steroid hormones). Moreover, we have added an extra paragraph in the Discussion section to make our study more complex, as follows:
L336-339
“As reviewed by Bar et al. many studies postulate about some correlation be-tween ERα and intracellular receptors such us AhR in context of TBC (REF, REF). Nonetheless, there are no empirical studies, which prove the afore-mentioned phe-nomenon in context of the TBC. However, Oour data show a significant increase in the Ahr mRNA expression in the spermatogenic cells after the exposure to TBC with E2.”
We have addressed all issues indicated in the review report and believe that the revised version can meet the journal publication requirements.
